# Few-Shot Audio-Visual Learning of Environment Acoustics

**Sagnik Majumder**[1]    **Changan Chen**[1,2*]   **Ziad Al-Halah**[1*]    **Kristen Grauman**[1,2]
[1]UT Austin    [2]Facebook AI Research

## Abstract

Room impulse response (RIR) functions capture how the surrounding physical environment transforms the sounds heard by a listener, with implications for various applications in AR, VR, and robotics. Whereas traditional methods to estimate RIRs assume dense geometry and/or sound measurements throughout the environment, we explore how to infer RIRs based on a sparse set of images and echoes observed in the space. Towards that goal, we introduce a transformer-based method that uses self-attention to build a rich acoustic context, then predicts RIRs of arbitrary query source-receiver locations through cross-attention. Additionally, we design a novel training objective that improves the match in the acoustic signature between the RIR predictions and the targets. In experiments using a state-of-the-art audio-visual simulator for 3D environments, we demonstrate that our method successfully generates arbitrary RIRs, outperforming state-of-the-art methods and—in a major departure from traditional methods—generalizing to novel environments in a few-shot manner. Project: `http://vision.cs.utexas.edu/projects/fs_rir`.

## 1   Introduction

Sound is central to our perceptual experience—people talk, doorbells ring, music plays, knives chop, dishwashers hum. A sound carries information not only about the *semantics* of its source, but also the *physical space* around it. For instance, compare listening to your favorite song in a big auditorium to hearing the same song in your cozy bedroom: the auditory experience changes drastically due to differences in the environment. On its way to our ears, the sound undergoes various acoustic phenomena: direct sound, early reflections, and late reverberations. Consequently, we hear spatial sound shaped by the environment's geometry, the materials of constituent surfaces and objects, and the relative locations of the sound source and the listener. These factors together comprise the *room impulse response* (RIR)—the transfer function that maps an original sound to the sound that meets our ears or microphones [58].

Learning to model RIRs would have far-reaching implications for augmented reality (AR), virtual reality (VR), and robotics. In AR/VR, a truly immersive experience demands that the user hear sounds that are *acoustically matched* with the surrounding augmented/virtual space [60]. For example, imagine a situation in which users of an AR/VR app, who are at different locations, are speaking to each other through telepresence and moving about during the conversation. In mobile robotics, an agent cognizant of environment acoustics could better solve important embodied tasks, like localizing sounds, navigating to a target sound, or separating out a target sound(s) of interest. In any such application, one must be able to anticipate the environment effects for arbitrarily positioned sources observed from arbitrary receiver poses.

Traditional approaches to model room acoustics require extensive access to the physical environment. They either assume a full 3D mesh of the space is available in order to simulate sound propagation

---

*Equal contribution

36th Conference on Neural Information Processing Systems (NeurIPS 2022).

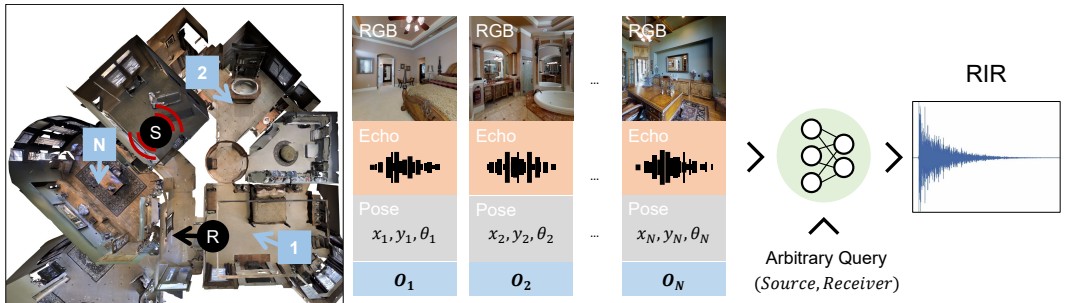

Figure 1: Given few-shot audio-visual observations from a 3D scene (blue boxes), we aim to learn an acoustic model for the entire environment such that we can generate a Room Impulse Response (RIR) for any arbitrary query of source (S) and receiver (R) locations in the scene—*without* observing images/echoes at those locations.

patterns [10, 42], or else require densely sampling sounds at many source-microphone position pairs all about the environment in order to measure the RIRs [26, 57]—both of which are expensive if not impractical. Recent work attempts to lighten these requirements by predicting (sometimes implicitly) the RIR from an image [55, 32, 23, 66, 62, 45, 9], but their output is specific to a single receiver position for which the photo exists, prohibiting generalization to other positions in the space.

Mindful of these limitations, we propose to infer RIRs in novel environments using only few-shot audio-visual observations. Motivated by how humans anticipate the overall structure of a 3D space by looking at a few parts of it, we hypothesize that imagery and echoes captured from a few different locations in a 3D scene can suggest its overall geometry and material composition, which in turn can facilitate interpolation of an RIR to arbitrary (unobserved) locations. See Figure 1.

To realize this idea, we propose a transformer-based model called FEW-SHOTRIR along with a novel training objective that facilitates high-quality prediction of RIRs by matching the energy decay between the predicted and the ground truth RIRs. FEW-SHOTRIR directly attends to the egocentric audio-visual observations to build an acoustic context of the environment. During training, our model learns the association between what is seen and heard in a variety of environments. Then, given a novel environment (e.g., a previously unseen multi-room home), the input is a sparse (few-shot) set of images together with the true RIRs at those image positions. In particular, each true RIR corresponds to positioning both the source and receiver where the image is captured, as obtained by emitting a short frequency sweep and recording the echoes. The output is an environment-specific function that can predict the RIR for arbitrary new source/receiver poses in that space—importantly, *without* traveling there or sampling any further images or echoes.

Our design has three key advantages: 1) the few-shot sparsity of the observations (on the order of tens, compared to the thousands that would be needed for dense visual or geometric coverage) means that RIR inference in a new space has low overhead; 2) the use of egocentric echoes means that the preliminary observations are simple to obtain, as opposed to repeatedly moving both a sound source (e.g., speaker) and the microphone independently to different relative positions; and 3) our novel differentiable training loss encourages predictions that capture the room acoustics, distinct from existing models that rely on non-differentiable RT60 losses [55, 50].

We evaluate FEW-SHOTRIR with realistic audio-visual simulations from SoundSpaces [10] comprising 83 real-world Matterport3D [7] environment scans. Our model successfully learns environment acoustics, outperforming the state-of-the-art models in addition to several baselines. We also demonstrate the impact on two downstream tasks that rely on the spatialization accuracy of RIRs: sound source localization and depth prediction. Our margin of improvement over a state-of-the-art model is as high as 23% on RIR prediction and 67% on downstream evaluation.

## 2   Related Work

**Audio Spatialization and Impulse Response Generation.**   Convolving an RIR with a waveform yields the sound of that source in the context of the surrounding physical space and the receiver location [4, 5, 19, 53]. Since traditional methods for measuring RIRs [26, 57] or simulating them

with sound propagation models [3, 10, 42] are computationally expensive, recent approaches indirectly generate RIRs by first estimating acoustic parameters [50, 16, 20, 31, 36, 61]—such as the reverberation time (RT60), the time an RIR takes to decay by 60 dB, and the direct-to-reverberant ratio (DRR), the energy ratio of direct and reflected sound—or matching the distributions of such acoustic parameters in real-world RIRs [48]. Whereas Fast-RIR [50] assumes that the environment size and reverberation characteristics are given, our model relies on learning directly from low-level multi-modal sensory information, thus generalizing to novel environments.

Alternately, some methods use images to predict RIRs of the target environment [55, 32] by implicitly inferring scene geometry [51] and acoustic parameters [32], or directly synthesizing spatial audio [23, 66, 62, 45, 9]. Although such image-based methods have the flexibility to extract diverse acoustic cues, their predictions are agnostic to the exact source and receiver locations, making them unsuitable for tasks where this mapping is important (e.g., sound source localization, audio goal navigation, or fine-grained acoustic matching).

Audio field coding approaches [46, 39, 47, 6] are able to model exact source and receiver locations, but to improve efficiency they rely on handcrafting features rather than learning them, which adversely impacts generation fidelity [35]. The recently proposed Neural Acoustic Fields (NAF) [35] tackles this by learning an implicit representation [41, 56] of the RIRs and additionally conditioning on geometrically-grounded learned embeddings. While NAF can generalize to unseen source-receiver pairs from the same environment, it requires training *one model per environment*. Consequently, NAF is unable to generalize to novel environments, and both its training time and model storage cost scale with the number of environments. On the contrary, given a few egocentric audio-visual observations from a novel environment, our model learns an implicit acoustic representation of the scene and predicts high-quality RIRs for arbitrary source-receiver pairs.

**Audio-Visual Learning.** Advances in audio-visual learning benefit many tasks, like audio-visual source separation and speech enhancement [1, 2, 13, 17, 27, 40, 43, 52, 64, 65, 37, 38], object/speaker localization [28, 29], and audio-visual navigation [10, 11, 8, 21, 15, 12, 63]. Using echo responses along with vision to learn a better spatial representation [22], infer depth [14], or predict the floor-plan [44] of a 3D environment has also been explored. In contrast, our model leverages the synergy of egocentric vision and echo responses to infer environment acoustics for predicting RIRs. Results show that both the visual and audio modalities play a vital role in our model training.

## 3 Few-Shot Learning of Environment Acoustics

We propose a novel task: few-shot audio-visual learning of environment acoustics. The objective is to predict RIRs on the basis of egocentric audio-visual observations captured in a 3D environment. In particular, for a few randomly drawn locations in the 3D scene, we are given egocentric RGB, depth images, and the echoes heard at those positions (from which the corresponding RIR can be computed, detailed below). Using those samples, we model the scene's acoustic space in order to predict RIRs for arbitrary pairings of sound source location and receiver pose (i.e., microphone location and orientation).

**Task Definition.** Specifically, let $\mathcal{O} = \{O_i\}^N$ be a set of $N$ observations ($N \leq 20$ in our experiments) randomly sampled from a 3D environment, such that $O_i = (V_i, A_i, P_i)$ where $V_i$ is the egocentric RGB-D view from a 90° field of view (FoV), $A_i$ is the RIR of the binaural echo response from a pose $P_i = (x_i, y_i, \theta_i)$ at location $(x_i, y_i)$ and orientation $\theta_i$. Given a query for an arbitrary source and receiver pair, $Q = (s_j, r_k)$, where $s_j = (x_j, y_j)$ is the omnidirectional sound source location and $r_k = (x_k, y_k, \theta_k)$ is the receiver microphone pose, which includes both its location and orientation, the goal is to predict the binaural RIR $\mathcal{R}^Q$ for the query $Q$. Thus, our goal is to learn a function $f$ to predict the RIR for an arbitrary query $Q$ given the egocentric audio-visual context $\{O_i\}^N$, such that $\mathcal{R}^Q = f(Q; \{O_i\}^N)$. Note that the query contains neither images nor echoes.

This task requires learning from both visual and audio cues. While the visual signal conveys information about the local scene geometry and the material composition of visible surfaces and objects, the audio signal, in the form of echo responses, is more long-range in nature and additionally carries cues about acoustic properties, the global geometric structure, and material distribution in the environment—beyond what's visible in the image. Our hypothesis is that sampling and aggregating

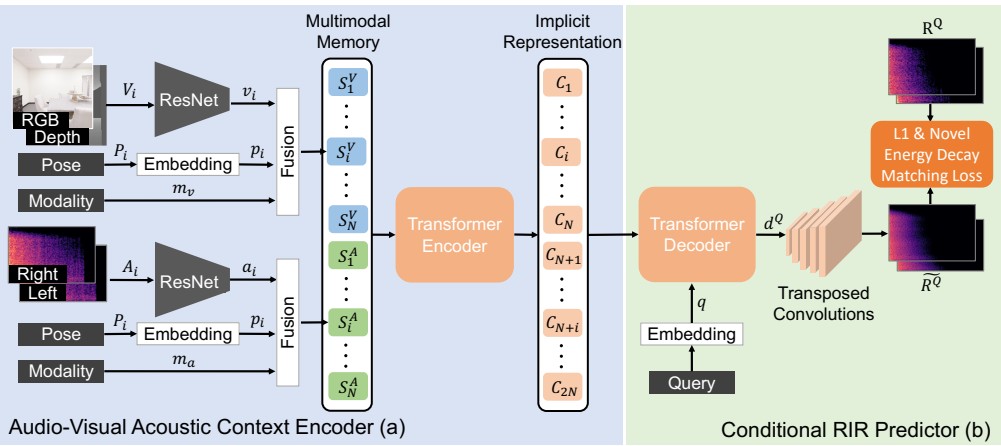

Figure 2: Our model predicts room impulse responses (RIR) for arbitrary source-receiver pairs in a 3D environment, including novel scenes, by building its implicit acoustic representation in the few-shot context of egocentric audio-visual observations. We train our model with a novel energy decay matching loss that helps capture desirable acoustic properties in its predictions.

these two complementary signals from a sparse set of locations in a 3D scene can facilitate inference of the full acoustic manifold and, consequently, enable high-quality prediction of arbitrary RIRs.

## 4  Approach

We introduce a novel approach called FEW-SHOTRIR for arbitrary RIR prediction in a 3D environment based on few-shot context of egocentric audio-visual observations. Our model has two main components (see Fig 2): 1) an audio-visual (AV) context encoder, and 2) a conditional RIR predictor. The AV context encoder builds an implicit model of the environment's acoustic properties by extracting multimodal cues from the input AV context (Sec.4.1). The RIR predictor uses this implicit representation of the scene and, conditioned on a query for an arbitrary source and receiver pair, it predicts the respective RIR (Sec.4.2)

Our model is trained end-to-end to reduce the error in the predicted RIR compared to the ground-truth using a novel training objective (Sec.4.3). Our objective not only encourages our model predictions to match the target RIRs at the spectrogram level, but also ensures that the predictions and the targets are similar with respect to important high-level acoustic parameters, thereby improving prediction quality. Next, we describe these two model components and the proposed training objective in detail.

### 4.1  Audio-Visual Context Encoder

Our AV context encoder (Fig. 2a) extracts features from the observations $\{O_i\}$. This context is sampled from the unmapped environment via an agent (a person or robot) traversing the scene and taking a small set of AV snapshots at random locations. Our model starts by embedding each observation $O_i = (V_i, A_i, P_i)$ using visual, acoustic, and pose networks. This is followed by a multi-layer transformer encoder [59] to learn an implicit representation of the scene's acoustic properties.

**Visual-Embedding.**  We encode the visual component $V_i$ by first normalizing its RGB and depth images such that all image pixels lie in the range $[0, 1]$. We then concatenate the images along the channel dimension and encode them with a network $f^V$ (a ResNet-18 [25]) into visual features $v_i$.

**Acoustic-Embedding.**  To measure the RIRs for the echo inputs, we use the standard sine-sweeping technique [18]. We first generate a "chirp" in the form of a sinusoidal sweep signal from 20Hz-20kHz (the human audible range) at the sound source, capture the resulting spatial sound with a microphone that has a (nearly) flat frequency response, and then retrieve the RIR at the receiver by convolving the spatial sound with the inverse of the sweep signal. We then use the short-time Fourier transform

(STFT) to represent all RIRs as magnitude spectrograms [55, 35] of size $2 \times F \times T$, where $F$ is the number of frequency bins, $T$ is the number of overlapping time windows, and each spectrogram has 2 channels. Finally, having converted the observed binaural echoes to an RIR $A_i$, we compute its log magnitude spectrogram and encode it with a network $f^A$ (a ResNet-18 [25]) into audio features $a_i$.

**Pose-Embedding.** To embed the camera pose $P_i$ into feature $p_i$, we first normalize all poses in $\{O_i\}$ to be relative to the first pose in the context $P_0$ and then represent each $P_i$ with a sinusoidal positional encoding [59].

**Modality-Embedding.** To enable our model to distinguish between the visual and audio modalities in the context, we introduce a modality token $m_i \in \{m_V, m_A\}$ such that the visual $m_V$ and acoustic $m_A$ modality embeddings are learned during training. While the visual modality (RGB-D images) reveals the local geometric and semantic structure of the environment, the acoustic modality (the echoes) carry more global acoustic information about regions of environments that are both within and out of the field of view. Our modality-based embedding allows our model to attend within and across modalities to capture both modality-specific and complementary environmental cues for a comprehensive modeling of the acoustic properties of the scene.

**Context Encoder.** For each visual observation in $\{O_i\}$ we concatenate its embedding $v_i$ with its pose $p_i$ and modality $m_V$ and project this representation with a single linear layer to get visual input $S_i^V$. Similarly, we concatenate $a_i$, $p_i$, and $m_A$ and project it with another linear layer to get the acoustic input $S_i^A$. This creates a multimodal memory of size $2N$, such that $S = \{S_0^V, \ldots, S_N^V, S_0^A, \ldots S_N^A\}$. Next, our context encoder attends to the embeddings in $S$ with self-attention to capture the short- and long-range correlations within and across modalities and through multiple layers to learn the implicit representation $C = \{C_1, \ldots, C_{2N}\}$ that models the acoustic properties of the 3D scene. This representation is then fed to the next module that generates the RIR for an arbitrary source-receiver query, as we describe next.

## 4.2 Conditional RIR Predictor

Given an arbitrary source-receiver query $Q = \{s_j, r_k\}$, we first normalize the poses of $s_j$ and $r_k$ relative to $P_0$ and encode each with a sinusoidal positional encoding as we did in Sec.4.1 to generate the pose encodings $p_j^s$ and $p_k^r$, respectively. Then, we concatenate and project $[p_j^s, p_k^r]$ using a single linear layer to get the query encoding $q$. Next, our RIR predictor, conditioned on $q$, performs cross-attention on the learned implicit representation $C$ using a transformer decoder [59], and generates an encoding $d^Q$ that is representative of the target RIR for query $Q$ (i.e., $R^Q$). Again, we stress that the query consists of only poses—no images or echoes.

We upsample $d^Q$ with transpose convolutions using a multi-layer network $U$ to predict the magnitude spectrogram in the log space for the RIR. Finally, we transform this log magnitude spectrogram back to the linear space to obtain our model's RIR prediction $\tilde{R}^Q$ for a query $Q$.

## 4.3 Model Training

Our model is optimized to predict the target RIR $R^Q$ for a given query $Q$ during training in a supervised manner using a loss $L$ that captures both the prediction accuracy of the spectrogram as well as high-level acoustic properties of the predicted $\tilde{R}^Q$ compared to the ground truth $R^Q$. Our loss $L$ contains two terms: 1) an $L_1$ reconstruction loss [55, 35] on the magnitude spectrogram of the RIR, and 2) a novel energy decay matching loss $L_D$.

For a target binaural spectrogram $R^Q$ with $F$ frequency levels and $T$ temporal windows, the $L_1$ loss tries to reduce the average prediction error in the time-frequency domain:

$$L_1 = \frac{1}{2 \times F \times T} \sum_{i=1}^{2 \times F \times T} ||\tilde{R}_i^Q - R_i^Q||_1.$$

On the other hand, the $L_D$ loss tries to capture the reverberation quality of the RIR by matching the temporal decay in energy of the predicted RIR with the target. $L_D$ allows our model to reduce errors in important reverberation parameters that depend on the energy decay of the RIR, like RT60,

which is the time taken by an impulse to decay by 60 dB, and DRR, which is the direct-to-reverberant energy ratio (cf. Table 1). Although past approaches [55, 50] have tried to minimize the RT60 error directly, incorporating an RT60 loss in a training objective is not viable due to the non-differentiable nature of the RT60 function. On the contrary, our proposed $L_D$ is completely differentiable and can be combined with any other RIR training objective.

To compute $L_D$, we first (similar to [55]) obtain the energy decay curve of an RIR by summing its spectrogram along the frequency axis to retrieve a full-band amplitude envelope, and then use Schroeder's backward integration algorithm to compute the decay curve $D_\mathcal{E}$. However, unlike [55], which computes the RT60 value from the decay curve through a series of non-differentiable operations, our $L_D$ directly measures the error in the energy decay curve between the prediction and the target, which not only makes it completely differentiable but also useful for capturing different energy based acoustic properties other than RT60, like DRR and early decay time (EDT) [49]. Towards that goal, we compute the absolute error in $D_\mathcal{E}$ between $\tilde{R}^Q$ and the target $R^Q$ for those temporal positions at which the target energy decay $D_\mathcal{E}(R^Q)$ is non-zero. This lets our model ignore optimizing for the all-zero tails in shorter RIRs. $L_D$ is defined as follows:

$$L_D = \frac{1}{2 \times T} \sum_{i=1}^{2 \times T} ||D_\mathcal{E}(\tilde{R}^Q)_i - D_\mathcal{E}(R^Q)_i||_1 \odot \mathbb{1}[D_\mathcal{E}(R^Q)_i \neq 0]$$

Our final training objective is $L = L_1 + \lambda L_D$, where $\lambda$ is the weight for $L_D$. We train our model using Adam [30] with a learning rate of $10^{-4}$ and $\lambda = 10^{-2}$.

## 5 Experiments

**Evaluation setup.** We evaluate our task using a state-of-the-art perceptually realistic 3D audio-visual simulator. In particular, we use the AI-Habitat simulator [54] with the SoundSpaces [10] audio and the Matterport3D scenes [7]. While Matterport3D contains dense 3D meshes and image scans of real-world houses and other indoor spaces, SoundSpaces provides pre-computed RIRs to render spatial audio at a spatial resolution of 1 meter for Matterport3D. These RIRs capture all major real-world acoustic phenomena (see [10] for details). This framework enables us to evaluate our task on a large number of environments, to test on diverse scene types, to compare methods under the same settings, and to report reproducible results. To our knowledge, there is no existing public dataset having both imagery and dense physically measured RIRs. Furthermore, due to the popularity of this framework (e.g., [22, 11, 15, 37, 44, 9]) we can test our model on important downstream tasks for RIR generation that are relevant to the larger community.

**Dataset splits.** We evaluate with 83 Matterport3D scenes, of which we treat 56 randomly sampled ones as *seen* and the remaining 27 as *unseen*. Unseen environments are only used for testing. For the seen environments, we hold out a subset of queries $Q$ for testing and use the rest for training and validation. Our test set consists of 14 sets of 50 arbitrary queries for each environment, where our model uses the same randomly chosen observation set for all queries in a set. This results in a train-val split with 8,107,904 queries, and a test split with 39,900 queries for *seen* and 18,200 queries for *unseen*. Our testing strategy allows us to evaluate a model on two different aspects: 1) for a given observation set from an environment, how the RIR prediction quality varies as function of the query, and 2) how well the model generalizes to environments previously unseen during training.

**Observations.** We render all RGB-D images for our model input at a resolution of $128 \times 128$ and sample binaural RIRs at a rate of 16 kHz. To generate the RIR spectrograms, we compute the STFT with a Hann window of 15.5 ms, hop length of 3.875 ms, and FFT size of 511. This results in two-channel spectrograms, where each channel has 256 frequency bins and 259 overlapping temporal windows. Unless otherwise specified, for both training and evaluation, we use egocentric observation sets of size $N = 20$ samples for our model.

**Existing methods and baselines.** We compare our approach to the following baselines and state-of-the-art methods (see Supp. for implementation and training details):

- **Nearest Neighbor:** a naive baseline that outputs the input echo's RIR that is closest to the query $Q$ in terms of the receiver pose $r$.

Table 1: RIR prediction results. All methods here train a single model to handle all seen/unseen environment queries. See Table 2 for comparisons with NAF [35], which trains one model per seen environment. All metrics use base $10^{-2}$ and lower is better. Results are statistically significant between our model and the nearest baselines ($p \leq 0.05$).

| Model | Seen environments | | | | Unseen environments | | | |
|---|---|---|---|---|---|---|---|---|
| | STFT | RTE | DRRE | MOSE | STFT | RTE | DRRE | MOSE |
| Nearest Neighbor | 4.65 | 1.15 | 385 | 24.4 | 4.87 | 1.26 | 391 | 28.0 |
| Linear Interpolation | 4.44 | 1.22 | 393 | 24.3 | 4.67 | 1.32 | 403 | 27.2 |
| AnalyticalRIR++ | 2.94 | 0.98 | 463 | 28.1 | 3.02 | 1.19 | 467 | 29.4 |
| Fast-RIR [50]++ | 1.37 | 1.25 | 137 | 13.7 | 1.45 | 1.61 | 369 | 15.2 |
| **FEW-SHOTRIR (Ours)** | **1.10** | **0.43** | **106** | **8.66** | **1.22** | **0.65** | **164** | **10.5** |
| Ours w/ $N = 1$ | 1.69 | 0.74 | 362 | 17.2 | 1.70 | 0.90 | 372 | 17.4 |
| Ours w/o echoes | 1.63 | 0.63 | 334 | 19.5 | 1.67 | 0.95 | 357 | 20.0 |
| Ours w/o vision | 1.63 | 0.67 | 332 | 19.2 | 1.58 | 0.83 | 347 | 19.3 |
| Ours w/o $L_D$ | 1.39 | 1.60 | 347 | 14.0 | 1.44 | 2.11 | 363 | 14.5 |

- **Linear Interpolation:** a naive baseline that computes the top four closest observation poses for the query receiver, and outputs the linear interpolation of the corresponding echoes' RIRs.
- **AnalyticalRIR++:** We modify our model to predict RT60 and DRR for the query using the egocentric observations. The modification uses the same transformer encoder-decoder pair, but replaces the transposed convolutions with fully-connected layers for RT60 or DRR prediction. It then *analytically* shapes an exponentially decaying white noise [33] on the basis of these two parameters to estimate the target RIR.
- **Fast-RIR [50]++:** Fast-RIR [50] is a state-of-the-art model that trains a GAN [24] to synthesize RIRs for rectangular rooms on the basis of environment and acoustic attributes, like scene size and RT60, which it assumes to be known a priori. Since FEW-SHOTRIR makes no such assumptions and is not restricted to rectangular rooms, we improve this method into Fast-RIR++: we use our modified model for AnalyticalRIR++ to estimate both the target RT60 and DRR, and use panoramic depth images at the query source and receiver to infer the scene size. We also train this model by augmenting the originally proposed objective with our $L_D$ loss to further improve its performance.
- **Neural Acoustic Fields (NAF) [35]:** a state-of-the-art model that uses an implicit scene representation [41] to model RIRs. As discussed above, a NAF model can only predict new RIRs in the same training scene; it cannot generalize to an unseen environment without retraining a new model from scratch to fit the new scene.

**Evaluation Metrics.** We consider four metrics: 1) **STFT** Error, which measures the average error between predicted and target RIRs at the spectrogram level; 2) **RT60 Error (RTE)** [55, 49, 50], which measures the error in the RT60 value for our predicted RIRs. 3) **DRR Error (DRRE)** [49], which measures the error in the estimated energy ratio between direct and reverberant sounds in an RIR; and 4) **Mean Opinion Score Error (MOSE)** [9], which uses a deep learning objective [34] to measure the difference in perceptual quality between a prediction and the target when convolved with human speech. While STFT Error measures the fine-grained agreement of a prediction to the target, RTE and DRRE capture the extent of acoustic mismatch in a prediction, and MOSE evaluates the level of perceptual realism for human speech.

## 5.1 RIR Prediction Results

Table 1 (top) reports our main results. The naive Nearest Neighbor and Linear Interpolation baselines incur very high STFT error, which shows that using echoes from poses that are spatially close to the query receiver as proxy predictions are insufficient, emphasizing the difficulty of the task. AnalyticalRIR++ fares better than the naive baselines on STFT and RTE but has a higher DRRE and MOSE, showing that reconstructing an RIR using simple waveform statistics is not enough. Fast-RIR++ shows the strongest performance among the baselines. Its improvement over AnalyticalRIR++, with which it shares the RT60 and DRR predictors, shows that high-quality RIR prediction benefits from learned methods that go beyond estimating simple acoustic parameters.

Table 2: RIR prediction results for our model vs. NAF [35]. All metrics use base $10^{-2}$.

| | Seen environments (3) | | | | | | | | Unseen environments (all) | | | |
| | Seen zones | | | | Unseen zones | | | | | | | |
| Model | STFT | RTE | DRRE | MOSE | STFT | RTE | DRRE | MOSE | STFT | RTE | DRRE | MOSE |
|---|---|---|---|---|---|---|---|---|---|---|---|---|
| NAF | 2.31 | 10.2 | 523 | 36.1 | 2.22 | 2.65 | 322 | 34.7 | 3.62 | 5.11 | 463 | 56.6 |
| **Ours** | **0.90** | **1.97** | **95.7** | **7.08** | **1.58** | **1.25** | **155** | **13.6** | **1.22** | **0.65** | **164** | **10.5** |

| Model | STFT | RTE | DRR | MOSE |
|---|---|---|---|---|
| Nearest Neighbor | 6.88 | 68.8 | 386 | 22.7 |
| Linear Interpolation | 6.61 | 68.7 | 388 | 21.5 |
| AnalyticalRIR++ | 3.37 | 7.95 | 474 | 26.0 |
| NAF [35] | 3.62 | 51.1 | 367 | 56.6 |
| Fast-RIR [50]++ | 1.58 | **1.23** | 436 | 17.1 |
| **FEW-SHOTRIR (Ours)** | **1.51** | 1.30 | **202** | **14.0** |

Table 3: RIR prediction results with ambient environment sounds in *unseen* environments. All metrics use base $10^{-2}$.

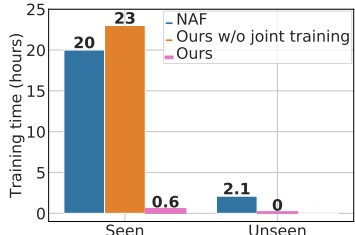

Figure 3: Training time comparison vs. NAF [35]

Our model outperforms all baselines by a statically significant margin ($p \leq 0.05$) on both *seen* and *unseen* environments. This shows that our approach facilitates environment acoustics modeling in a way that generalizes to novel environments without any retraining. Furthermore, its performance improvement over Fast-RIR [50]++ emphasizes the advantage of directly predicting RIRs on the basis of the implicit acoustic model inferred from egocentric observations, as opposed to indirect synthesis of RIRs by first estimating high-level acoustic characteristics for the target. As expected, our model has more limited success for very far-field queries; widely separated source and receivers make modeling late reverb difficult (see Supp for details).

**FEW-SHOTRIR (Ours) vs. NAF [35].**    Recall that, unlike our approach, NAF requires training one model per environment. Thus, for fair comparison, we train one model per scene for both NAF and our method. Due to the high computational cost of training NAF (as we discuss later) we limit our training to three large *seen* environments. Further, we split each seen environment into *seen zones*, which we use for training and also testing the models' interpolation capabilities, and *unseen zones*, which test intra-scene generalization. To give NAF access to our model's observations, we finetune it on our model's echo inputs before testing. For *unseen* environments, our model adopts the setup from the previous section, whereas we train NAF from scratch on our model's observed echoes; note that NAF's scene-specificity does not allow finetuning of a model trained on a seen environment.

Table 2 shows the results. Our model significantly ($p \leq 0.05$) outperforms NAF [35] on seen environments when considering both *seen* and *unseen zones*. The *seen zone* results underscore the better interpolation capabilities of our model in comparison to NAF when tested on held-out queries from the training zones. Our method's improvement over NAF on *unseen zones* shows that our model design and training objective lead to much better intra-scene generalization, even when NAF is separately finetuned on the observation sets from the *unseen zones*. While NAF improves over the naive baselines from Table 1 on STFT error, it does worse than other methods on most metrics. This demonstrates that just learning to predict a limited amount of echoes contained in the observation set of our model is insufficient to accurately model acoustics for unseen environments.

Figure 3 compares the training cost between NAF [35] and our model using wall clock time, when both models are trained on 8 NVIDIA Quadro RTX 6000 GPUs. When we train one model per environment, NAF takes 20 hours to converge on average, while our model takes 23 hours. However, our model design allows us to train one model jointly on all 56 Matterport3D training scenes in 32 hours, which reduces the average training time by $50\times$—down to 0.6 hours per environment. Moreover, for unseen environments, training NAF on echoes requires 2.1 hours for each observation set. In contrast, our model design enables training on a large number of scenes at a much lower average cost, while also allowing generalization to novel environments without further training.

| Model | Seen | | Unseen | |
|---|---|---|---|---|
| | SLE | DPE | SLE | DPE |
| True RIR (*Upper bound*) | 14.9 | 0.97 | 17.0 | 1.25 |
| Nearest Neighbor | 202 | 1.50 | 214 | 1.57 |
| Linear Interpolation | 202 | 1.39 | 213 | 1.49 |
| AnalyticalRIR++ | 254 | 1.64 | 270 | 1.69 |
| NAF [35] | – | – | 329 | 1.68 |
| Fast-RIR [50]++ | 168 | 1.39 | 201 | 1.52 |
| **FEW-SHOTRIR (Ours)** | **50.3** | **1.35** | **64.6** | **1.45** |

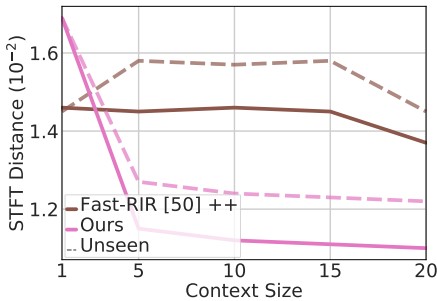

Figure 4: STFT error vs. context size $N$.

Table 4: Downstream task evaluation of RIR predictions for sound source localization and depth estimation. All metrics use a base of $10^{-2}$ and lower is better.

## 5.2 Model Analysis

**Ablations.** In Table 1 (bottom) we ablate the components of our model. When removing one of the modalities from the input, we see a drop in performance, which indicates that our model leverages the complementatry information from both vision and audio to learn a better implicit model of the scene. We also observe a performance drop across all metrics, especially on the RTE metric, upon removing our energy decay matching loss $L_D$. This shows that having $L_D$ as part of the training objective allows our model to better capture desirable reverberation characteristics for the target query, like RT60, while also additionally helping to score better on other metrics. Furthermore, we see that reducing the observation set to just 1 sample, i.e., $N = 1$, impacts our model's performance. However, even under this extreme condition, our model still shows better generalization compared to several baselines. We further investigate the impact of context size on our model's performance in Figure 4. Our model already reduces the error significantly using a context size of 5, with diminishing reductions as it gets a larger context. This plot also highlights the low-shot success of our model vs. the strongest baseline, Fast-RIR++.

**Ambient environment sounds.** We test our model's ability to generalize in the presence of ambient and background sounds. To that end, we repeat the experiment from Sec.5.1 where this time we insert a random ambient or background sound (e.g., running heater, dripping water). This background noise impacts the echoes input to our model. Table 3 reports the results. Even in this more challenging setting, our method substantially improves over all methods on almost all metrics.

**Qualitative results.** Figure 5 shows two RIR prediction scenarios for our model: *high reverberation*, where the query receiver is located close to the source in a very narrow and reverberant corridor surrounded by walls, and *low reverberation*, where the source and receiver are spread apart in a more open space. Our model shares the same observation samples across these settings. With high reverb, our model focuses on vision due to its ability to better reveal the compact geometry of the surroundings and its effects on scene acoustics, whereas echoes are distorted from strong reverberation. For low reverb, echoes are probably more informative about the acoustics of the more open surroundings due to their long-range nature. However, in both cases, our model prioritizes samples that provide good coverage of the overall scene, rather than just scoping out the local area around the query. This allows our model to make predictions that closely match the targets.

## 5.3 Downstream Applications for RIR Inference: Source Localization and Depth Estimation

Next we consider two downstream tasks inspired by AR/VR and robotics: sound source localization and depth estimation from echoes. For both tasks, spatial audio generated by more accurate RIRs should translate into better representation of the true acoustics, and hence better downstream results.

We train a model for each task using ground truth RIR magnitude spectrograms and evaluate using the predicted magnitude spectrograms from our model and all baselines for the same set of queries from both *seen* and *unseen* environments. Table 4 reports the results. DPE is the average $L_1$ error between a normalized depth target and its prediction, and SLE is the average $L_1$ error in prediction

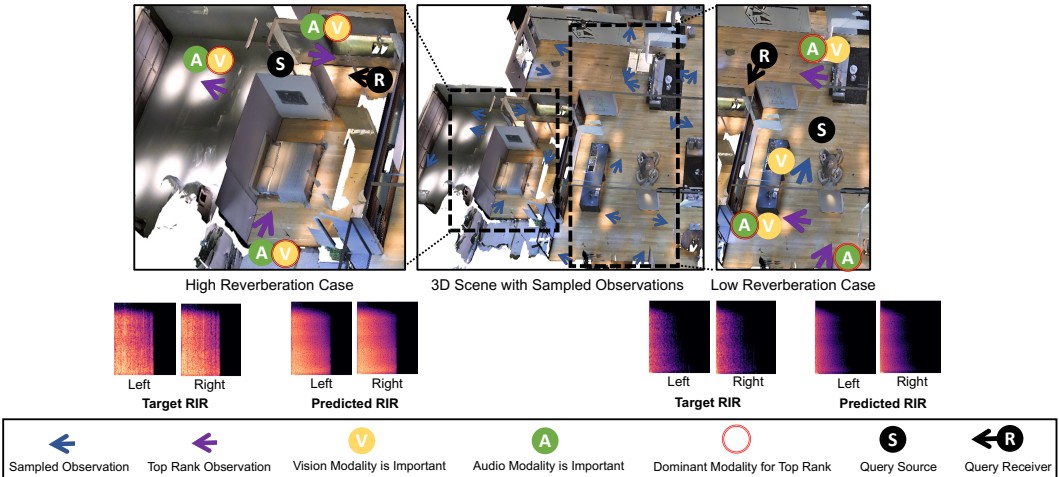

Figure 5: RIR predictions for a high and low reverberation case, where our model uses the same observations in both cases. For high reverb, our model relies more on vision than echoes for inferring scene acoustics, since echoes could be misleading in this case due to its long reverberation tail. For low reverb, our model uses echoes more, likely because their long-range nature better informs about the acoustics of the more open surroundings.

of the source location (in meters) relative to the receiver for a query. Our method outperforms all baselines by a statistically significant margin ($p \leq 0.05$). In particular, for the more difficult *unseen* environments, our model reduces the error relative to the ground truth upper bound by 74% for SLE and 25% for DPE when compared to Fast-RIR++, highlighting that our model's predictions capture spatial and directional cues more precisely than all other baselines and existing methods.

## 6   Potential Societal Impact

Our model enables modeling the acoustics in a 3D scene using only few observations. This has multiple applications with a positive impact. For example, accurate modeling of the scene acoustics enables a robot to locate a sounding object more efficiently (like finding a crying baby, or locating a broken vase). Additionally, this allows for a truly immersive experience for the user in augmented and virtual reality applications. However, RIR generative models allow the user to match the acoustic reverberation in their speech to an arbitrary scene type, and hence hide their true location from the receiver, which may have both positive and negative implications. Finally, our model uses visual samples from the environment for more accurate modeling of the acoustic properties of the scene. However, the dataset used in our experiments contains mainly indoor spaces that are of western designs, and with a certain object distribution that is common to such spaces. This may bias models trained on such data toward similar types of scenes and reduce generalization to scenes from other cultures. More innovations in the model design to handle strong shifts in scene layout and object distribtutions, as well as more diverse datasets are needed to mitigate the impact of such possible biases.

## 7   Conclusion

We introduced a model to infer arbitrary RIRs having observed only a small number of echoes and images in the space. Our approach helps tackle key challenges in modeling acoustics from limited observations, generalizing to unseen environments without retraining, and enforcing desired acoustic properties in the predicted RIRs. The results show its promise: substantial gains over existing models, faster training, and benefits for downstream source localization and depth estimation. In future work, we plan to explore ways to optimize the placement of the observation set and explore ways to curate large-scale real world data for sim2real transfer.

**Acknowledgements:** Thanks to Tushar Nagarajan and Kumar Ashutosh for feedback on paper drafts. UT Austin is supported in part by the IFML NSF AI Institute, NSF CCRI, and DARPA L2M. K.G. is paid as a research scientist by Meta, and C.C. was a visiting student researcher at Facebook AI Research when this work was done.

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
