# Few-Shot Audio-Visual Learning of Environment Acoustics Supplementary Material

**Sagnik Majumder**[1]    **Changan Chen**[1,2*]   **Ziad Al-Halah**[1*]      **Kristen Grauman**[1,2]

[1]UT Austin    [2]Facebook AI Research

In this supplementary material we provide additional details about:

- Video (with audio) for qualitative illustration of our task and qualitative evaluation of our model predictions (Sec. 1).

- Evaluation of the impact of the query source location on our model's prediction quality for a fixed receiver (Sec. 2).

- Audio dataset details (Sec. 3), as mentioned in Sec. 5 and checklist item 3b of the main paper.

- Model architecture details for RIR prediction (Sec. 4.1) and downstream tasks (Sec. 4.2), as noted in Sec. 5 of the main paper.

- Training hyperparameters (Sec. 4.3), as referenced in Sec. 5 and checklist item 3b of the main paper.

## 1   Supplementary Video

The supplementary video, available at `https://vision.cs.utexas.edu/projects/fs_rir/`, shows the perceptually realistic SoundSpaces [2] audio simulation platform that we use for our experiments, and provides a qualitative illustration of our task, Few-Shot Audio-Visual Learning of Environment Acoustics. Moreover, we qualitatively demonstrate our model's prediction quality by comparing the predictions with the ground truths, both at the RIR level and in terms of perceptual similarity when the RIRs are convolved with real-world monaural sounds, like speech and music. We also analyze common failure cases for our model (Sec. 5.1 in main) and qualitatively show how our model predictions can be used to successfully localize an audio source in a 3D environment. Please use headphones to hear the spatial audio correctly.

## 2   Impact of the Source Location on the Prediction Error

In Fig. 1, we show the RIR prediction error as a function of different source locations for a fixed receiver location. As we can see, the prediction error tends to be small when the source is relatively close to the receiver, or there are no major obstacles along the path connecting them. This indicates that the model leverages the local geometry of the scene and the acoustic information captured from echoes for better predictions. However, the error increases when there are large distances between the source and receiver (Sec. 5.1 in main), and especially when there are major obstacles for audio propagation in between (e.g., walls, narrow corridors). Modeling how audio gets transformed on such a long path becomes very challenging due to the limited observations available to the model and the larger scene area that contributes to transforming the audio.

---

*Equal contribution

36th Conference on Neural Information Processing Systems (NeurIPS 2022).

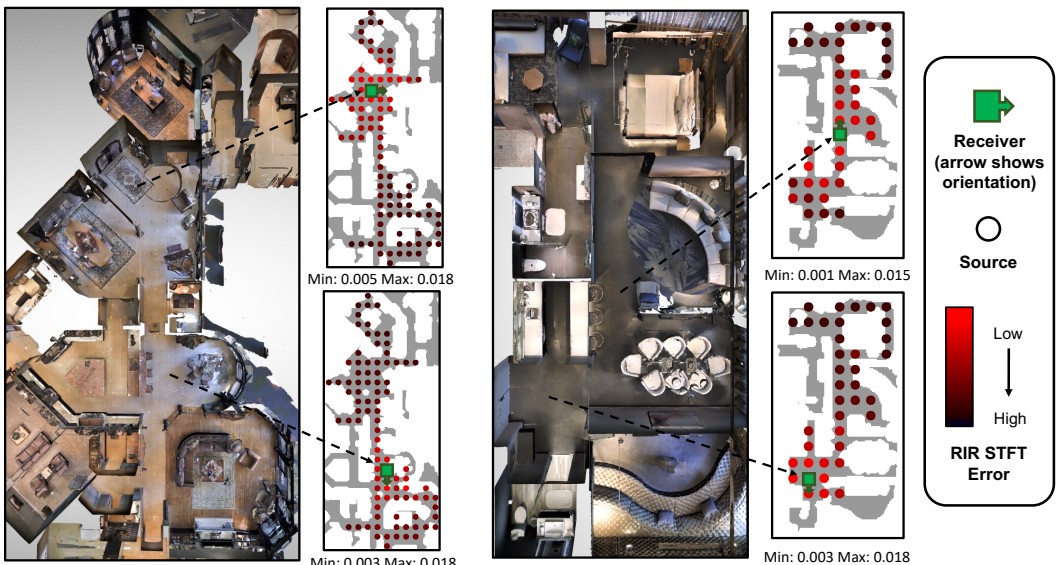

Figure 1: RIR prediction STFT error as a function of varying source locations (filled circles) for a given receiver (a green square with an arrow). We show two scenes and two examples per scene. The color of the circle at the source location indicates the STFT error in the RIR prediction associated with that source and receiver pair. The error in each example is normalized between the min and max values shown underneath the map.

# 3 Audio Dataset

For computing the mean opinion score error (MOSE) [1] (Sec. 5 in main), we sample 5 second long speech clips from the LibriSpeech [9] dataset, which comprise both male and female speakers. For every test query, we randomly choose one of the sampled clips and convolve with the true RIR or a model's prediction for that query to estimate the corresponding mean opinion score (MOS) [7] and, subsequently, the error in MOS for a model's prediction relative to the true RIR. We use a 5-second-long temporal window for all model predictions and true RIRs when estimating their MOS.

For our experiment with ambient environment sounds (Sec. 5.2 in main), we use ambient sounds from the ESC-50 [10] dataset (e.g., dog barking, running water). For every test query, we randomly sample a location in the 3D scene for an ambient sound and play a randomly chosen 1 second long clip from the ESC-50 dataset at that location. To retrieve the observed binaural echo response $A_i$ (Sec. 3 and Sec. 4.1 in main) in this setting, we first convolve the clean echo RIR for each observation $O_i$ with the sinusoidal sweep sound, then mix it with the binaural the ambient sound for its pose $P_i$, and finally deconvolve using the inverse sweep (Sec. 4.1 in main).

We will publish the link to our datasets on our project page.

# 4 Architecture and Training

Here, we provide our architecture and additional training details for reproducibility. We will publish the link to our codebase on our project page.

## 4.1 Model Architectures for RIR Prediction

**Visual Encoder.** Our visual encoder $f^V$ is a ResNet-18 [4] model (Sec. 4.1 in main) that takes egocentric RGB and depth images from the observation set, which are concatenated channel-wise, as input and produces a 512-dimensional feature.

**Acoustic Encoder.** Our acoustic encoder $f^A$ is another ResNet-18 [4] (Sec. 4.1 in main) that separately encodes the binaural log magnitude spectrogram for an echo RIR $A_i$ into a 512-dimensional feature.

**Pose Encoder.** To embed an observation pose $P_i$ or a query source-receiver pair $Q$ (Sec. 3 in main), we use sinusoidal positional encodings [14] (Sec. 4.1-2 in main) with 8 frequencies, which generate a 16-dimensional feature vector (the positional encodings comprise both sine and cosine components with 8 features per component) for every attribute of an observation pose or a query (i.e., $x$, $y$, and $\theta$).

**Modality Encoder.** For our modality embedding $m$ (Sec. 4.1 in main) we maintain a sparse lookup table of 8-dimensional learnable embeddings, which we index with 0 to retrieve the visual modality embedding ($m_V$) and 1 to retrieve the acoustic modality embedding ($m_A$).

**Fusion Layer.** To generate the multimodal memory $S$ (Sec. 4.1 in main) for our context encoder (Sec. 4.1 in main), we separately concatenate the modality features (produced by $f^V$ for vision and $f^A$ for echo responses) for an observation, the corresponding sinusoidal pose embedding, and the modality embedding ($m_V$ for visual features and $m_A$ for acoustic features), and project using a single linear layer to 1024-dimensional embedding space. Similarly, to generate the query encoding $q$ for our conditional RIR predictor (Sec. 4.2 in main), we use another linear layer to project the query's sinusoidal positional encodings to a 1024-dimensional feature vector. Furthermore, we don't use bias in any fusion layer.

**Context Encoder.** Our context encoder (Sec. 4.2 in main) is a transformer encoder [14] with 6 layers, 8 attention heads, a hidden size of 2048 and ReLU [13, 8] activations. Additionally, we use a dropout [12] of 0.1 in our context encoder.

**Conditional RIR Predictor.** Our conditional RIR predictor (Sec. 4.2 in main) has 2 components: 1) a transformer decoder [14] to perform cross-attention on the implicit representation $C$ (Sec. 4.1 in main), which is produced by the previously described context encoder, using the query encoding $q$ (Sec. 4.2 in main), and 2) a multi-layer transpose convolution network $U$ (Sec. 4.2 in main) to upsample the decoder output $d^Q$ and predict the magnitude spectrogram for the query $Q$ in log space.

The transformer decoder [14] has the same architecture as our context encoder.

The transpose convolution network $U$ comprises 7 layers in total. The first 6 layers are transpose convolutions with a kernel size of 4, stride of 2, input padding of 1, ReLU [13, 8] activations and BatchNorm [5]. The number of input channels for the transpose convolutions are 128, 512, 256, 128, 64 and 32, respectively. The last layer of $U$ is a convolution layer with a kernel size of 3, stride of 1, padding of 1 along the height dimension and 2 along the width dimension, and 16 input channels. Finally, we switch off bias in all layers of $U$.

## 4.2 Model Architectures for Downstream Tasks

**Sound Source Localization.** We use a ResNet-18 [4] feature encoder that takes the log magnitude spectrogram of an RIR (predicted or ground truth as input). We take the encoded features and feed them to a single linear layer that predicts the location coordinates of a query's source relative to the query's receiver pose.

**Depth Estimation.** Following VisualEchoes [3], we use a U-net [11] that takes the log magnitude spectrogram of an echo as input and predicts the depth map (Sec. 5.3 in main) as seen from the echo's pose. The encoder of our U-net has 6 layers. The first layer is a convolution with a kernel size of 3, stride of 2, padding of 1 along the height dimension, and 2 input channels. The remaining 5 layers are convolutions with a kernel size of 4, padding of 1 and stride of 2. These 5 layers have 64, 64, 128, 256 and 512 input channels, respectively. Each convolution is followed by a ReLU activation [13, 8] and a BatchNorm [5].

The decoder of the U-net has 5 transpose convolution layers. Each transpose convolution has a kernel size of 4, stride of 2 and input padding of 1. Except for the last layer that uses a sigmoid activation function to generate depth maps, which are normalized such that all pixels are in the range of $[0, 1]$, each transpose convolution has a ReLU activation [13, 8] and a BatchNorm [5]. The decoder layers have 512, 1024, 512, 256 and 128 channels, respectively. We use skip connections between the encoder and the decoder starting with their second layer. We switch off bias in both the encoder and decoder.

### 4.3 Training Hyperparameters

In addition to the training details specified in main (Sec. 4.3), we use a batch size of 24 during training. Furthermore, for every entry of the batch, we query our model with 60 arbitrary source-receiver pairs for the same observation set, which effectively increases the batch size further and improves training speed. Other training hyperparameters specific to our Adam [6] optimizer include $\beta_1 = 0.9$, $\beta_2 = 0.999$ and $\epsilon = 10^{-5}$.