# OpenReview forum: "Few-Shot Audio-Visual Learning of Environment Acoustics"
_NeurIPS.cc/2022/Conference — NeurIPS 2022 Accept_

### Official Review · Reviewer_WWei · 2022-07-06

**Rating:** 8
**Confidence:** 3
**Soundness:** 4 excellent
**Presentation:** 3 good
**Contribution:** 3 good

**Summary:**

This paper presents a new method for RIR prediction given previously recorded RIRs and images for different locations in the scene. The authors propose a new architecture which fuses audio and visual encoders, followed by a transformer and a decoder. They use two loss functions, L1 and a novel Decay Loss which solves the non-differentiability of RT60.

**Questions:**

Can the authors explain the applicability of this precise method? Is there a specific setup where this way of solving the problem is optimal compared to image -> RIR, specifically where you might need to model source/emitter positions with such fine grained accuracy? A more detailed discussion of the advantages of this method would be useful in the paper.

**Limitations:**

The authors briefly mention the collection of RIRs during inference by playing a short sweep. I think this process should be addressed in more detail, for example do you need a flat frequency response microphone (or does it generalize), how does the loudness of the emitter factor in, etc. It's quite likely that collecting these ground truth RIRs is difficult, especially if you don't have access to the environment at a perfectly quiet moment. This means that making this work in more realistic scenarios such as public places may not be reasonable.

**Strengths And Weaknesses:**

Overall the paper is quite good. It is well written, and explains the motivation of different design choices and experiments. The included video was quite nice, especially the section on limitations. It showed the performance compared to ground truth, including a real-world environment which was not simulated. The experiments show comparisons with other baselines, such as nearest neighbors which is an important baseline. The novel loss for RT60 is a very welcome contribution, since L1 loss is suboptimal when trying to model reverb decay.

My main concern is that the problem formulation is not widely applicable and may not be too useful. In general, asking the user to capture ground truth RIR in multiple locations of an environment is difficult, and requiring images at those locations also make it hard. The authors claim that an advantage is not needing images at the new location for inferring RIR, but collecting an image at the new location is often easier than collecting ground truth RIR and images in multiple locations. Given that there are many papers which do image -> RIR, the contribution of this work is somewhat limited. The main application I could imagine is for robots navigating a space which have the ability to collect ground truth RIR.

---

> ### Author Response · Authors · 2022-08-02
> **Authors’ response to Reviewer WWei [Part 1]**
>
> We thank the reviewer for the very encouraging comments and insightful questions. We provide a two-part response.
> \
> \
> \
> **Q1. My main concern is that the problem formulation is not widely applicable and may not be too useful. In general, asking the user to capture ground truth RIR in multiple locations of an environment is difficult, and requiring images at those locations also make it hard.**
>
> In our few-shot setup, only a few samples (as few as 5 samples) are needed from the scene to predict RIRs of a reasonable quality (Fig. 5 and L327-30). The scenario we envision is where an AR user or a mobile robot briefly introduces the system to a new environment by pausing in a few locations (L141-3), after which point the system can generalize high-quality RIRs for any other positions and for the many (unobserved) source receiver pairings. In contrast, traditional state-of-the-art methods for measuring RIRs involve either densely sampling sounds at many source receiver pairs all around the environment [1, 2] or simulating sound propagation by capturing the complete environment mesh using thousands of observations [3, 4].
>
> To use our approach, a user just has to capture images and the echo responses at few sampling locations. The RIRs can then be analytically extracted from the captured echo responses (see L150-3). In other words, collecting the ground truth RIR amounts to emitting a sound and recording the echo response.
>
> For our experiment with a real-world scene (Supp. video 5:31 - 6:52), we collected the samples by mounting a camera, speaker, and microphone pair on a stick, which was straightforward and took ∼3 minutes per sample on average. While we made our own simple rig for this purpose, the increasing availability of lightweight smart glasses equipped with cameras, speakers, and microphones (e.g., Ray Ban Stories) would make this process even simpler.
> \
> \
> \
> **Q2. The authors claim that an advantage is not needing images at the new location for inferring RIR, but collecting an image at the new location is often easier than collecting ground truth RIR and images in multiple locations. Given that there are many papers which do image -> RIR, the contribution of this work is somewhat limited.**
>
> First, we stress that an image->RIR model assumes access to images that our model does not. Our model uses a sparse context of few samples somewhere in the space (as few as 5) but *neither visual nor acoustic data is sampled for the query RIR*. That is, if we want to predict the RIR at 100 different locations in the scene, our model uses 0 samples from these locations while an image->RIR method will use 100 images (L37-9).
>
> In terms of efficiency, we also have the advantage of generalizing to any new receiver position. Collecting images at every new receiver location would be expensive, especially when the receiver is continuously moving in the environment (e.g., an AR app user moving in a room while on a video call). On the contrary, our approach collects a sparse set of AV samples only once for every new environment and then can predict the RIRs for arbitrary source-receiver pairs.
>
> In terms of accuracy, images alone can be insufficient for precise modeling of the environment acoustics, as they always do not reveal the full geometry and material composition of the environment. For example, we provide the Fast-RIR [5]++ baseline with acoustic properties (RT60 and DRR) along with panoramic images (L259-60) to infer room size, and yet it is outperformed by our model. Whereas existing image-based methods do not precisely model source and receiver locations (L83) and are often restricted to one source-receiver pair for which the image has been captured (L38-9), we can predict RIRs for any source-receiver pair in a new environment accounting for the specific camera/mic poses.
> \
> \
> \
> \
> **Response references**
>
> [1] Martin Holters, Tobias Corbach, and Udo Zölzer. "Impulse response measurement techniques and their applicability in the real world". 2009.
>
> [2] Guy-Bart Stan, Jean-Jacques Embrechts, and Dominique Archambeau. "Comparison of different impulse response measurement techniques". Journal of the Audio Engineering Society, 50(4):249–262, April 2002.
>
> [3] Changan Chen, Unnat Jain, Carl Schissler, Sebastia Vicenc Amengual Gari, Ziad Al-Halah, Vamsi Krishna Ithapu, Philip Robinson, and Kristen Grauman. "Soundspaces: Audio-visual navigation in 3d environments". In European Conference on Computer Vision, pages 17–36. Springer, 2020.
>
> [4] Damian Murphy, Antti Kelloniemi, Jack Mullen, and Simon Shelley. "Acoustic modeling using the digital waveguide mesh". IEEE Signal Processing Magazine, 24(2):55–66, 2007.
>
> [5] Anton Ratnarajah, Shi-Xiong Zhang, Meng Yu, Zhenyu Tang, Dinesh Manocha, and Dong Yu. "Fast-rir: Fast neural diffuse room impulse response generator". In IEEE International Conference on Acoustics, Speech and Signal Processing (ICASSP), pages 571–575. IEEE, 2022.

---

> > ### Author Response · Authors · 2022-08-02
> > **Authors’ response to Reviewer WWei [Part 2]**
> >
> > **Q3. Can the authors explain the applicability of this precise method? Is there a specific setup where this way of solving the problem is optimal compared to image -> RIR, specifically where you might need to model source/emitter positions with such fine grained accuracy? A more detailed discussion of the advantages of this method would be useful in the paper.**
> >
> > Our method is useful for AR/VR applications that aim to provide a real-time and immersive experience to its users by generating acoustically matched sounds at different locations in an environment (L26-8). For example, imagine a situation in which users of an AR/VR app, who are at different locations, are speaking to each other through telepresence and moving about continuously during the conversation. In addition, our work can also be used to better build mobile robots for solving tasks like audio goal navigation, audio source separation, and audio source localization (L29-32). All these applications need fine-grained modeling of environment acoustics for arbitrary source and receiver locations. We will make the applications and advantages of our method more clear in the paper.
> > \
> > \
> > \
> > **Q4. The authors briefly mention the collection of RIRs during inference by playing a short sweep. I think this process should be addressed in more detail, for example do you need a flat frequency response microphone (or does it generalize)**
> >
> > The microphone used to measure the RIRs in the SoundSpaces \[3\] dataset has a flat frequency response. However, we noticed that our model generalizes to a real-world scenario reasonably well (Supp. video 5:31-6:52), where the echo responses are captured with a [Zylia](https://www.zylia.co/white-paper.html) microphone that doesn’t have a perfectly flat frequency response. Also, note that the frequency response behavior is dependent on the microphone hardware and can be used as a configurable parameter to customize the simulation’s frequency response for better generalization.
> > \
> > \
> > \
> > **Q5. how does the loudness of the emitter factor in, etc. It’s quite likely that collecting these ground truth RIRs is difficult, especially if you don’t have access to the environment at a perfectly quiet moment. This means that making this work in more realistic scenarios such as public places may not be reasonable.**
> >
> > Note that we use the SineSweep technique for computing the echo response RIRs (L150-3), which is a widely-accepted standard [1, 2, 6] for accurately measuring RIRs in the real world. In noisy environments, the loudness of the emitted sweep signal can affect the quality of the measured RIRs for the echo responses and consequently, the RIR prediction accuracy for query source-receiver pairs. To test our model’s robustness to such noise, in the submitted paper we evaluated it in the presence of ambient environment sounds from the ESC-50 [7] dataset. We observe that 1) all models show lower performance compared to the non-noisy setting, and 2) our model outperforms the baselines on almost all metrics (Table 3 and L331-5). Moreover, to test our model’s robustness to varying degrees of noise loudness, we varied the ambient noise level so that the mean SNR of the echo responses ranges from 0.5 dB to 20 dB. Our model’s performance remains reasonably stable – the relative change in RIR STFT error is as low as 2%. Finally, when we collected the data for our experiment with a real-world scene (Supp. video 5:31 - 6:52), the environment had ambient environment sounds, like door closing, running AC, etc. Nevertheless, our model was able to predict the RIRs for the query locations reasonably well.
> > \
> > \
> > \
> > \
> > **Response references**
> >
> > [6] Angelo Farina. "Simultaneous measurement of impulse response and distortion with a swept-sine technique". Journal of the Audio Engineering Society, February 2000.
> >
> > [7] Karol J. Piczak. "ESC: Dataset for Environmental Sound Classification". In Proceedings of the 23rd Annual ACM Conference on Multimedia, pages 1015–1018. ACM Press.

---

### Official Review · Reviewer_4JYb · 2022-07-09

**Rating:** 6
**Confidence:** 4
**Soundness:** 3 good
**Presentation:** 3 good
**Contribution:** 3 good

**Summary:**

A typical way to render 3D audio simulating sounds emanating from specific directions is by means of convolving location-agnostic clean audios with binaural room impulse responses (BRIRs). This is useful for:
1. AR/VR applications
2. Augmentations for training Neural networks which generalize well across different acoustic conditions.
In this paper, the authors come up with a novel approach to incorporate the contextual information necessary for estimating RIRs in unseen environments for arbitrary placement of source and receiver. Essentially the idea is to use audio-visual information from an ego-centric  90 deg field-of-view (FOV) camera, RIR from a specified source location assuming receiver (microphone) is at the origin from N arbitrarily sampled spatial locations to create a contextual embedding. In the test phase while generating the required RIR for a specified source and receiver location, This contextual embedding is used with a query embedding (which incorporates the information about source and receiver locations) to generate an RIR-embedding which is dilated to generate Binaural RIR magnitude spectra.

**Questions:**

1. Why is the proposed approach expected to work in unseen environments ? More precisely say the network has always been trained on images from enclosures with smooth reflecting walls. How would this network learn the impact of rough walls where scattering is pre-dominant ?
2. What in the proposed approach helps the network learn difference in reflecting properties of different surfaces ?

**Limitations:**

1. Visually similar looking surfaces with different reflection properties could be challenging for this proposed approach leading to unsuitable RIRs ? This is definitely not a societal impact but a technical limitation to consider and address.

**Strengths And Weaknesses:**

Strengths:
1. The idea is very interesting.
2. the paper is well written easy to understand.
3. The experiments are indeed insightful with relevant baselines. The authors explanation of results for each experiments is appropriate and concise.

Weakness:
1. It is not clear how the authors convert the Binaural magnitude spectrogram of RIRs to waveform for construct signals for downstream source localization tasks. This conversion in itself could affect some RIR properties and it is not clear how this is addressed in this work.

---

> ### Author Response · Authors · 2022-08-02
> **Authors’ response to Reviewer 4JYb**
>
> We thank the reviewer for the encouraging feedback and insightful questions.
> \
> \
> \
> **Q1. It is not clear how the authors convert the Binaural magnitude spectrogram of RIRs to waveform for constructing signals for downstream source localization tasks. This conversion in itself could affect some RIR properties and it is not clear how this is addressed in this work.**
>
> To clarify, we do not convert them to waveforms. Our goal with the source localization task is to gauge how precisely the predicted RIRs capture directional cues. Towards that goal, we directly use the RIRs [1, 2, 3] and feed their magnitude spectrograms [4, 5] (Supp. L102-3) to a CNN for source localization (i.e., without converting to the waveform).
> \
> \
> \
> **Q2. Why is the proposed approach expected to work in unseen environments? More precisely say the network has always been trained on images from enclosures with smooth reflecting walls. How would this network learn the impact of rough walls where scattering is pre-dominant?**
>
> Our training data comprises 56 large and diverse Matterport3D [6] scenes (L229-30), which capture different possible configurations of objects and materials common in real-world environments. We evaluate our model on the remaining 27 unseen Matterport3D environments (L229-30) and an unseen real-world scene (Supp. video 5:31-6:52), testing our model’s ability to generalize to previously unseen geometric and material configurations. Further, our model leverages the sparse context sampled from the current scene to learn about its acoustic and material properties and generalize to new locations in the same environment. As in any machine learning problem, handling entirely out-of-distribution data (such as entirely unobserved surface types) could be challenging.
> \
> \
> \
> **Q3. What in the proposed approach helps the network learn difference in reflecting properties of different surfaces? Visually similar looking surfaces with different reflection properties could be challenging for this proposed approach leading to unsuitable RIRs ? This is definitely not a societal impact but a technical limitation to consider and address.**
>
> The synergy of the visual (RGB-D image) and audio (echo response) inputs—and our model’s ability to aggregate the information contained in these two signals from different sampling locations using the transformer-based design—allows it to reason about reflective properties of different surfaces. In cases where similar looking surfaces have different reflective properties, the audio cue can help the model to distinguish between such surfaces. Without either vision or audio input, we see a large drop in RIR prediction quality (Table 1 and L318-21).
> \
> \
> \
> \
> **Response references**
>
> [1] Daniele Salvati, Carlo Drioli, and Gian Luca Foresti. "Sound source and microphone localization from acoustic impulse responses". IEEE Signal Processing Letters, 23(10):1459–1463, 2016.
>
> [2] Reza Parhizkar, Ivan Dokmani ́c, and Martin Vetterli. "Single-channel indoor microphone localization". In 2014 IEEE International Conference on Acoustics, Speech and Signal Processing (ICASSP), pages 1434–1438, 2014.
>
> [3] Mohammad Javad Taghizadeh, Afsaneh Asaei, Saeid Haghighatshoar, Philip N. Garner, and Hervé Bourlard. "Spatial sound localization via multipath euclidean distance matrix recovery". IEEE Journal of Selected Topics in Signal Processing, 9(5):802–814, 2015.
>
> [4] Nelson Yalta, Kazuhiro Nakadai, and Tetsuya Ogata. "Sound source localization using deep learning models". Journal of Robotics and Mechatronics, 29:37–48, 02 2017.
>
> [5] Toni Hirvonen. "Classification of Spatial Audio Location and Content using Convolutional Neural Networks". Journal of the Audio Engineering Society, May 2015.
>
> [6] Angel Chang, Angela Dai, Thomas Funkhouser, Maciej Halber, Matthias Niessner, Manolis Savva, Shuran Song, Andy Zeng, and Yinda Zhang. "Matterport3d: Learning from rgb-d data in indoor environments". arXiv preprint arXiv:1709.06158, 2017.

---

### Official Review · Reviewer_hzCe · 2022-07-09

**Rating:** 5
**Confidence:** 3
**Soundness:** 3 good
**Presentation:** 3 good
**Contribution:** 3 good

**Summary:**

This paper introduces a transformer-based method that uses self-attention to build a rich acoustic context, then predicts RIRs of arbitrary query source-receiver locations through cross-attention. Additionally, a novel training objective is proposed to improve the match in the acoustic signature between the RIR predictions and the targets. Experiments show that the proposed method outperforms prior works.

**Questions:**

See the weaknesses

**Limitations:**

The limitations and negative impact are discussed in the supplement.

**Strengths And Weaknesses:**

Strengths

- This paper is well written and easy to read
- The performance is good.
- The training objective is novel

Weaknesses

- The transformer-based model is lack of novelty as it just contains normal architecture design with attention.
- The claimed "few-shot learning" is confusing, as the method uses a lot of data for training. I understand that there are a few observations for each sample so it is called "few-shot" in the paper. But I am not sure whether it is suitable to call the task "few-shot".

---

> ### Author Response · Authors · 2022-08-02
> **Authors’ response to Reviewer hzCe**
>
> We thank the reviewer for the insightful feedback and comments.
> \
> \
> \
> **Q1. The transformer-based model is lack of novelty as it just contains normal architecture design with attention.**
>
> We do not claim the transformer architecture as a technical contribution. The key novelty of our work is to build a rich acoustic context of a 3D environment on the basis of a sparse set of audio-visual observations, thereby enabling RIR prediction for arbitrary query source-receiver locations (L6-8 and L45-8). In contrast, prior work requires extensive access to the physical environment (L33-39). To our knowledge, we are also the first to use a transformer-based approach in a few-shot and multimodal setting for the task of RIR prediction (L45-62), as also noted by the other 3 reviewers. We also introduce a novel energy decay matching loss $L_D$ (L189-215), which contributes significantly to the RIR prediction quality of our model (Table 1 and L321-4).
> \
> \
> \
> **Q2. The claimed "few-shot learning" is confusing, as the method uses a lot of data for training. I understand that there are a few observations for each sample so it is called "few-shot" in the paper. But I am not sure whether it is suitable to call the task "few-shot".**
>
> Few-shot learning is used to describe our approach’s ability to learn an implicit acoustic model of a novel environment (i.e., not seen during training) using few audio-visual samples from the space, (L47-58). Our use of the "few-shot" term is consistent with the common use in the literature. For example, in few-shot object classification [1, 2, 3, 4], the model has access to a large dataset of images during training (e.g., ImageNet [5]), but then the model builds an object classifier using only a few images of the novel category. We will make sure to further clarify this terminology in the text.
> \
> \
> \
> \
> **Response references**
>
> [1] Yaqing Wang, Quanming Yao, James T Kwok, and Lionel M Ni. "Generalizing from a few examples: A survey on few-shot learning". ACM computing surveys (csur), 53(3):1–34, 2020.
>
> [2] Michael Fink. "Object classification from a single example utilizing class relevance metrics". In Advances in Neural Information Processing Systems, volume 17. MIT Press, 2004.
>
> [3] Li Fei-Fei, R. Fergus, and P. Perona. "One-shot learning of object categories".
> IEEE Transactions on Pattern Analysis and Machine Intelligence, 28(4):594–611, 2006.
>
> [4] Oriol Vinyals, Charles Blundell, Timothy Lillicrap, Daan Wierstra, et al. "Matching networks for one shot learning". Advances in neural information processing systems, 29, 2016.
>
> [5] Jia Deng, Wei Dong, Richard Socher, Li-Jia Li, Kai Li, and Li Fei-Fei. "Imagenet: A large-scale hierarchical image database". In 2009 IEEE Conference on Computer Vision and Pattern Recognition, pages 248–255, 2009.

---

### Official Review · Reviewer_gkRK · 2022-07-11

**Rating:** 8
**Confidence:** 3
**Soundness:** 4 excellent
**Presentation:** 4 excellent
**Contribution:** 3 good

**Summary:**

This paper presents a novel approach to predict Room Impulse Responses in a few-shot learning setting, where only very limited data is available. More specifically, only a small number of echoes obervation in the room along with RGB pictures of the room. The proposed model is a multi-modal transformer-based model, taking the RIR of a given location or an image as input, along with a pose encoding and a modality encoding. The model is trained with two losses, a fairly standard L1 loss and a novel energy decay loss which help the model focus on the most important part of the RIR. The model is trained and evaluated using synthetic data and compared with baseline and state-of-the-art models. The proposed model is shown to yield better perfomance on seen and unseen rooms.

**Questions:**

- The hyper-parameters of the transformer-based Encoder and Decoder are not documented anywhere (or I missed them), could the authors provide them?


**Limitations:**

The limitations are not discussed in the paper, the authors should discuss the issues of real life application, typically the data could be noisy, either the echo audio measurement (street noise) or the pictures (bad lightening, occlusion, etc).

**Strengths And Weaknesses:**

Strengths:
- The approach is entirely novel as far as I can tell.
- The experiments are sounds and clearly show the benefit on the proposed approach. The ablation study is a nice addition.
- The results are significant, as the proposed model yield better performance than state-of-the-art models, even one being published this year.
- The paper is clearly written and well structured.

Weaknesses:
- The only drawback is that the experiments are done on synthetic data, not real measurements. But as the authors discussed, there seems to be no available dataset for that.

Overall the approach is very neat, very interesting, completely novel as far as I can tell and quite significant. Even if the field is very specific (RIR), I think the paper is worth publishing at NeurIPS as the approach is interesting and could be applied to other tasks.

---

> ### Author Response · Authors · 2022-08-02
> **Authors’ response to Reviewer gkRK**
>
> We thank the reviewer for the very encouraging comments and feedback.
> \
> \
> \
> **Q1. The only drawback is that the experiments are done on synthetic data, not real  measurements. But as the authors discussed, there seems to be no available dataset for  that.**
>
> In our experiments, we reduce the simulation to real gap by using photo-realistic 3D scenes (Matterport3D [1]) and state-of-the-art sound simulation (SoundSpaces [2]). This allows us to do large-scale evaluation in a realistic audio-visual setup (L218-28).
>
> As the reviewer notes, there is no existing public dataset with real-world visual and acoustic RIR measurements. However, in the submission we did qualitatively test our approach in a real-world environment by collecting observations and queries from it. See Supp. video (5:31 - 6:52). We observe that our model predictions are perceptually similar to the ground-truth, even in this real-world setting. They also provide an accurate sense of the source locations. We plan to build on these initial findings in future work by collecting large-scale real-world data (L365-6).
> \
> \
> \
> **Q2. The hyper-parameters of the transformer-based Encoder and Decoder are not documented anywhere (or I missed them), could the authors provide them?**
>
> We provide the details of the model architecture and the training hyperparameters in Supp. Sec. 5 (L60-125) and L215-6. Additionally, we will release the code and data upon acceptance.
> \
> \
> \
> **Q3. The limitations are not discussed in the paper, the authors should discuss the issues of real life application, typically the data could be noisy, either the echo audio measurement (street noise) or the pictures (bad lighting, occlusion, etc.)**
>
> We discuss common failure cases for our model in the Supp. video (6:55 - 8:19), and in L288-90. We also discuss the potential negative implications of our work in Supp. Sec 2 (L26-34).
>
> We evaluate our model in the presence of ambient background sounds and noise, which do impact the echoes input to our model (L331-5). The noise audio sampled from the ESC-50 [3] dataset (Supp. L52-8) contains sounds that are common in street noise (e.g., people laughing, car horn, chain saw, rain, siren, train, etc.). Even in this challenging setting, our model outperforms the baselines on almost all metrics (Table 3). That said, the absolute accuracy declines for all methods in the presence of excessive noise, suggesting an important area for continued work.
>
> While the AI-Habitat simulator does not provide control on changing the lighting conditions in the scene, we tested our model in the extreme case of no visual input, i.e., completely dark environment, (Table 1 and L318-21) and observe that our model outperforms most baselines on all performance metrics. Additionally, we qualitatively evaluate our model with different lighting conditions from the Matterport3D [1] scenes using a real-world environment (Supp. video 5:31 - 6:52).
>
> We are not sure what type of occlusions the reviewer is referring to. However, our evaluation setup comprises scans from real-world environments, which capture natural household geometries and distributions of objects. Consequently, our setup includes occlusions commonly found in real-world scenes.
> \
> \
> \
> \
> **Response references**
>
> [1] Angel Chang, Angela Dai, Thomas Funkhouser, Maciej Halber, Matthias Niessner, Manolis Savva, Shuran Song, Andy Zeng, and Yinda Zhang. "Matterport3d: Learning from rgb-d data in indoor environments". arXiv preprint arXiv:1709.06158, 2017.
>
> [2] Changan Chen, Unnat Jain, Carl Schissler, Sebastia Vicenc Amengual Gari, Ziad Al-Halah, Vamsi Krishna Ithapu, Philip Robinson, and Kristen Grauman. "Soundspaces: Audio-visual navigation in 3d environments". In European Conference on Computer Vision, pages 17–36. Springer, 2020.
>
> [3] Karol J. Piczak. "ESC: Dataset for Environmental Sound Classification". In Proceedings of the 23rd Annual ACM Conference on Multimedia, pages 1015–1018. ACM Press.

---

### Author Response · Authors · 2022-08-02
**Overall response**

We thank the reviewers for their valuable feedback and suggestions. All reviewers recommended accepting the paper and liked several features of our work:
* idea is very interesting [R-4JYb]
* approach is novel and interesting [R-gkRK, R-4JYb, R-WWei]
* approach can be applied to other tasks [R-gkRK]
* training objective is novel and a welcome contribution [R-gkRK, R-hzCe, R-WWei]
* experiments are sound and insightful [R-gkRK, R-4JYb]
* results are strong and significant [R-gkRK, R-hzCe]
* ablation study is a nice addition [R-gkRK]
* paper is good, clearly written, well-structured and provides good and concise explanations [R-gkRK, R-hzCe, R-4JYb, R-WWei]
* quite nice supplementary video [R-WWei]
* addresses limitations [R-WWei]

Next, we respond to each reviewer individually.

---

### Meta-Review · Area_Chair_tQuF · 2022-08-23

**Recommendation:** Accept
**Confidence:** Certain

**Metareview:**

Reviewers are in agreement that the paper should be accepted, and the authors were able to address concerns leading to an increase in score from one of the reviewers.

**Award:**

No

---

### Decision · Program_Chairs · 2022-09-14

Accept